# Ecological and Evolutionary Processes Shaping Viral Genetic Diversity

**DOI:** 10.3390/v11030220

**Published:** 2019-03-05

**Authors:** Cas Retel, Hanna Märkle, Lutz Becks, Philine G. D. Feulner

**Affiliations:** 1Department of Fish Ecology and Evolution, Center for Ecology, Evolution and Biogeochemistry, EAWAG, Swiss Federal Institute of Aquatic Science and Technology, 6047 Kastanienbaum, Switzerland; philine.feulner@eawag.ch; 2Division of Aquatic Ecology, Institute of Ecology and Evolution, University of Bern, 3012 Bern, Switzerland; 3Section of Population Genetics, TUM School of Life Sciences Weihenstephan, Technical University of Munich, 85354 Freising, Germany; hanna.maerkle@tum.de; 4Limnological Institute University Konstanz, Aquatic Ecology and Evolution, 78464 Konstanz, Germany; lutz.becks@uni-konstanz.de

**Keywords:** genetic diversity, viral population genetics, host–virus coevolution, eco-evolutionary feedback

## Abstract

The contemporary genomic diversity of viruses is a result of the continuous and dynamic interaction of past ecological and evolutionary processes. Thus, genome sequences of viruses can be a valuable source of information about these processes. In this review, we first describe the relevant processes shaping viral genomic variation, with a focus on the role of host–virus coevolution and its potential to give rise to eco-evolutionary feedback loops. We further give a brief overview of available methodology designed to extract information about these processes from genomic data. Short generation times and small genomes make viruses ideal model systems to study the joint effect of complex coevolutionary and eco-evolutionary interactions on genetic evolution. This complexity, together with the diverse array of lifetime and reproductive strategies in viruses ask for extensions of existing inference methods, for example by integrating multiple information sources. Such integration can broaden the applicability of genetic inference methods and thus further improve our understanding of the role viruses play in biological communities.

## 1. Introduction

Viruses are ubiquitous and diverse [1,2], adapt rapidly [3], and often engage in intimate relationships with their host [4]. Delimitation and discovery of new viruses has been strongly eased by the advent of new sequencing technologies, and genetic barcoding is nowadays a standard technique in virology [5,6,7]. The contemporary genome sequence and genetic variation within a given virus strain is shaped by the ongoing interaction of dynamic ecological and evolutionary forces. Thus, besides species delimitation, sequence data can provide valuable information about the evolutionary history of viruses and have been used to track transmission patterns [8], to date the emergence of new viruses [9,10] and host shifts [11], and to identify genes that are under selection [12]. However, extracting such information requires an understanding of all other factors which can (interactively) affect genetic change and create, maintain, and/or deplete genetic variation.

Our aim is to review the relevant evolutionary and ecological processes that affect viral genetic diversity and outline how different processes interact and temporally vary (or not). In this review we focus on antagonistic coevolutionary and eco-evolutionary feedback interactions between host and virus species, without restricting ourselves to a particular group of viruses. The influence of abiotic factors such as temperature, CO_2_ concentration, and UV radiation on viruses and the interaction with the host has been reviewed elsewhere [13,14]. We will furthermore give a brief overview of existing methodology designed to infer aspects of evolutionary history based on genetic data and describe some recently developed tools for the simultaneous analysis of host and virus genetic data. 

## 2. Viral Population Genetics

Variation at the sequence level of viruses is created by mutations which range from changes at single bases (single-nucleotide polymorphism, SNP) up to rearrangements of the genome architecture. Variant forms of a genetic sequence are called **alleles**, and the position at which they occur is referred to as **locus**. Viral mutation rate estimates range from 10^−8^ to 10^−6^ changes per base pair per cellular infection (~generation) for DNA viruses and from 10^−6^ to 10^−4^ for RNA viruses [15,16,17]. These rates are high compared to microbes such as *E. coli* and *S. saccharomyces* (both < 10^−9^ [18,19]). The total mutation supply in a population per generation not only depends on the mutation rate per sequence per generation (*μ*) but also on the effective population size *N_e_* (see below) of the focal population. The **population mutation rate**
*θ = 4N_e_μ* captures this interplay and represents the expected number of accumulated differences between a pair of randomly chosen sequences in a population [20]. The ultimate fate of a mutation, i.e., fixation, loss, or maintenance at intermediate frequency—and by extension the total amount of genomic variation in a population, is determined by the interaction between genetic drift, selection, recombination, and migration. In this review, we pay less attention to viral recombination [21,22] and the concepts of spatial structure and migration [23,24] but focus on drift and selection because they are especially relevant for microbial viruses. 

**Genetic drift** describes the process of stochastic changes in allele frequencies due to random sampling of offspring from the parental generation. Generally, the strength of genetic drift depends mainly on the effective population size, with smaller populations experiencing stronger drift. The **effective population size** (*N_e_*) corresponds to the size of an idealized population (satisfying the assumptions of the so-called Wright–Fisher model of population genetics: constant population size, non-overlapping generations, diploid individuals, equal sex ratio, no selection, no recombination, small variance in offspring numbers, and random mating among individuals) which experiences the same amount of stochastic genetic change as the population analyzed [25]. The ratio of *N_e_* to census population size *N* is affected by factors such as the mode of reproduction and temporal variation in population size [26]. Viruses possess several characteristics that reduce the *N_e_*-to-*N* ratio. Population sizes of viruses infecting several globally important phytoplankton species can fluctuate by orders of magnitude within a season [27,28,29,30]. Viruses typically also have skewed offspring distributions, with a lot of virions never successfully reproducing and a few contributing disproportionately large amounts of genetic material to the next generation [31]. For example, the RNA virus vesicular stomatitis virus and the dsDNA virus chlorovirus PBCV-1 can produce burst sizes ranging from 50 to 8000 and 100 to 350 particles per replication event, respectively [32,33]. Both fluctuating population size and skewed offspring distributions increase the relative importance of drift. Hence, viruses experience stronger drift than other organisms with similar census population sizes. 

Besides genetic drift, the type and strength of selection influences the probability and rate by which alleles increase or decrease in frequency in a population. The term **fitness** captures the number of offspring any individual possessing a particular genotype is expected to contribute to the next generation. **Positive selection** describes selection on constantly beneficial alleles [34], which are expected to increase in frequency across generations until they reach fixation, meaning that every individual in the population possesses the allele and variation at the locus is lost. Opposed to positive selection, **purifying selection** captures the process of selection against deleterious mutations. **Balancing selection** summarizes any form of selection which maintains variation in the population (i.e., more than one allele at a locus) [35]. 

Alleles under positive selection can decrease in frequency due to genetic drift. Therefore, there is always a chance that they are lost from a population, especially when their frequency is low (Figure 1). In a Wright–Fisher type population, the probability of fixation of a beneficial mutation present in a single individual, provided that it has a weak selective advantage *s* and population size is large, is approximately *2s N_e_/N* [36,37]. Skewed offspring distributions as seen in many viruses increase the probability that beneficial mutations reach fixation [38,39] and decrease the expected time this takes [40]. For these reasons, we expect frequency changes of alleles under selection in virus populations to be comparatively rapid. 

Alleles that have no or small fitness effects which are physically associated with (linked to) a positively selected mutation are expected to change simultaneously in frequency with the mutation under selection, a process referred to as **genetic hitchhiking** [41,42,43]. Being “dragged along” with alleles under positive selection increases the variance in temporal frequency changes of genetic hitchhikers compared to those of neutrally evolving loci [44,45]. Genetic diversity and frequency changes at neutral sites can be further affected by purifying selection against deleterious mutations, a process termed **background selection [46]**. Background selection decreases genetic variation at linked sites [47,48] and has the potential to slow down or even impede the expected frequency increase of linked adaptive alleles [49]. Associations between physically linked sites can be broken up by **recombination**. In the absence of recombination, two beneficial mutations that arise independently in different lineages will never be combined in a single genotype [50,51]. Rather, there will be competition among the offspring of these two lineages: a process termed **clonal interference [52,53]**. Clonal interference can result in hampered frequency increases and the extinction of one of the two lineages [45,54]. 

The combination of high mutation rate and large population size leads to a high supply of de novo mutations in virus populations. This increases the likelihood that multiple mutations with varying effects on fitness segregate simultaneously in virus populations [55,56], and both clonal interference and genetic hitchhiking are then likely to occur [57]. However, as viral genomes are in most cases rather small (but see recently discovered giant viruses, e.g., [58]) and densely packed with protein-coding regions, most mutations are likely to be highly deleterious. This suggests a prominent role for background selection in most viruses [39] and decreases the potential for multiple mutations to segregate simultaneously (but see [59], where genetic variation was found at hundreds of sites in human cytomegalovirus populations within host individuals). There is currently not enough empirical data available to make general statements about the occurrence and relative strength of interference between deleterious and beneficial mutations in viruses [60]. 

In summary, the observed variation at the genomic level results from a complex interplay between mutation supply, drift, and selection, and their individual contributions depend on the biology of the particular virus. 

## 3. Host–Virus Coevolution

Because viruses depend on their hosts for replication, their genome evolution is also strongly influenced by their host [61,62]. Similarly, hosts are under constant pressure to reduce the detrimental fitness effects of viruses [63,64]. This reciprocal evolutionary interplay is called coevolution when adaptation of one species changes selection on the interacting partner and vice versa [65,66,67]. The dynamics and genetic consequences of host–virus coevolution in particular and of antagonistic species interactions in general have been enigmatic research topics dedicated to understanding the maintenance of genetic diversity [68], the evolution of sex [69], patterns of local adaptation [70], and the speed of evolution [71]. 

An important determinant of host–virus coevolutionary dynamics is the number of genotypes per population and how these interact with antagonist genotypes, captured in an **infection matrix** [72]. The two opposite ends of the continuum of possible infection matrices are, on the one hand, **matching alleles**, where every virus genotype can only infect one host genotype [73,74] (Figure 2a), and, on the other, **gene-for-gene** interactions (Figure 2b), where virus genotypes infect a broad range of host genotypes [75]. There is a range of possible infection matrices in between these two extremes [76,77] (Figure 2c). Viruses infecting bacterial hosts completely span this range [78,79]. Importantly, adaptation of one or both interacting partners can lead to changes in the underlying infection matrix and in the resulting coevolutionary dynamics [80,81,82]. 

The coevolutionary dynamics of genes underlying the molecular interaction between host and virus are separated into **arms race** or **fluctuating selection** dynamics (also called trench-warfare or Red Queen dynamics) [83] although these two categories rather describe the two end points of a continuum [72]. In arms race dynamics, coevolution is driven by the reciprocal consecutive increases in frequency of novel genotypes which provide an evolutionary advantage (e.g., the ability to target a novel outer cell membrane protein [84]), ultimately resulting in fixation. On the genomic level, these frequency increases result in a reduction of genetic diversity at linked loci [41,43], also referred to as a selective sweep. Arms race dynamics are characterized by directional changes in phenotype distributions, such as a monotonic increase in viral infectivity [80,85,86]. Known examples of arms races in (semi-)natural species interactions are *Flavobacterium* phage coevolution in fish farms [87] and *Drosophila* resistance to sigma virus [88]. 

Fluctuating selection dynamics, a form of balancing selection, occur when the fitness of multiple functional genotypes in both species negatively depends on their frequency in the population. This results in fluctuations of their relative frequencies and thus maintenance of several genotypes in both interacting species. The underlying mechanism, here outlined for two host and two virus genotypes, is as follows: selection in the virus favors efficient exploitation of the most common host genotype *A*. This confers a selective advantage to a rarer host genotype *B*, which is thus expected to increase in frequency over the course of generations. Once host genotype *B* becomes the most common one, selection in the virus no longer favors exploitation of host genotype *A*, granting another virus capable of infecting host genotype *B* the selective advantage. Such frequency-dependent selection patterns can lead to perpetual oscillations of functional allele frequencies in both coevolving populations, with allele frequencies in the virus population following those in the host [89]. Fluctuating selection dynamics are characterized by oscillating phenotype distributions [90,91] and are expected to result in higher levels of genomic variation at functional and associated loci than expected under neutrality. Negative frequency-dependent host–parasite interactions have been found in a taxonomically wide range of antagonistically coevolving systems, such as flax and its fungal pathogen flax rust [92], *Daphnia magna* and its bacterial endoparasite *Pasteuria ramosa* [93], and *Pseudomonas* with naturally associated lytic viruses [94]. 

In summary, coevolutionary dynamics at the functional loci can be classified based on their effect on the number and frequencies of functional genotypes in both antagonistic species over time. The occurring type of dynamics depends on various factors, such as the number of functional genes, the underlying infection matrix of different genotypes, and the ecology of the interacting species. The effect on the genomic diversity at the interacting functional loci is determined by the interaction of the particular dynamics with de novo mutations, standing genetic variation, recombination, and the amount of genetic drift.

## 4. Eco-Evolutionary Feedbacks in Viruses

Host–virus systems are likely to be subject to feedbacks between ecological and evolutionary change. Population densities affect encounter rates between antagonistic individuals. Therefore, the strength of antagonistic selection varies in concert with population size. Population sizes, in turn, mediate the strength of drift and supply of de novo mutations (Section 2). For these reasons, abundance (ecological) and allele frequency (evolutionary) dynamics often reciprocally influence each other [95] in host–virus interactions [96] and interactively determine coevolutionary genetic change (Figure 3) [97,98]. 

Reciprocal effects between ecological and evolutionary change are especially important to consider when evolutionary and ecological changes occur on similar timescales, i.e., when evolution is rapid [99,100]. How often contemporary evolution has a considerable influence on community dynamics is an outstanding question in ecology and evolution [101,102]. Rapid evolution of resistance has been experimentally shown to change the effects chlorovirus has on the population dynamics of its host [82] and to facilitate coexistence with a third species [103]. Furthermore, rapid host resistance evolution has been demonstrated to alter the effects myovirus has on marine microbial food web structure [104]. With their short generation times and large population sizes, both viruses and microbes are likely to display rapid adaptive responses, and hence to be involved in eco-evolutionary feedbacks. 

Analytical predictions on allele frequency dynamics and equilibrium states change when population sizes are allowed to vary and determine the strength of antagonistic selection [105,106]. Mechanistic predictions on reciprocal allele frequency changes can then be done by combining information on phenotypic traits and abundances of both populations [107,108]. Balancing selection—thus, maintenance of higher-than-expected levels of genetic diversity—becomes more likely and can occur even when the infection matrix conforms to a ‘true’ gene-for-gene system (see Figure 2b), where virus genotypes are equally successful on all host genotypes [109]. However, population bottlenecks (drastic reductions in size) increase the probability that stochastic fixation events occur [110]. Such events remove functional genotypes and subsequently diminish genomic variation. Even if no genotypes are stochastically lost, the traditionally predicted simple harmonic oscillations of allele frequencies are either replaced by more complex combinations of sinusoidal functions [111], or allele frequencies stabilize but species abundances fluctuate [112,113] in models including eco-evolutionary feedback effects. 

In summary, strong reciprocal fitness effects that cause fluctuations in population size and the potential to adapt rapidly make microbial host–virus interactions likely subject to eco-evolutionary feedback dynamics. To which extent the integration of such feedback is necessary to correctly interpret the genetic signature of coevolution is an unresolved question in evolutionary genetics. 

## 5. Genomic Inference Methods

After having outlined the various processes which can affect and interact with viral genomic diversity, we will now give an introductory overview of available inference methods that can be used to extract information about these processes from genomic data. We will start with methods which are traditionally used to analyze genomic data of a single species. Then, we present some recently developed methods which take into account the reciprocal nature of host–virus coevolutionary interactions.

Outlier scans can be used to search for loci that are putatively under selection (Figure 4a). Genomes from a population sample are scanned for loci which show either elevated or decreased levels of genetic diversity and/or linkage disequilibrium compared to the genome-wide average. Deviations from the average are interpreted as evidence for selection having acted [34,114,115,116]. Depending on the type of available data, these scans are often based on summaries of the sequence data, such as the site frequency spectrum (from which statistics such as Tajima’s D can be calculated), haplotype distribution, or when multiple populations are compared, e.g., by differentiation measures such as *F*_ST_. 

The first step in outlier scanning is the establishment of the demographic history of the population based on diversity patterns of putatively neutrally evolving loci. This step is crucial, as various demographic scenarios can produce genomic signatures which are very similar to those of positive selection (recent population expansion after a bottleneck) or of balancing selection (population decline) [117,118,119]. It is further important to note that not accounting for background selection can result in biased demographic inference, most pronounced when it is at intermediate levels [48,117]. The second step involves comparing the diversity per locus to the expected neutral distribution given the established population demography. Loci under positive selection are expected to show lower levels of diversity and higher linkage disequilibrium with neighboring regions. Loci under balancing selection will on the other hand show elevated levels of nucleotide diversity, detection of which can be hard depending on the timescale at which selection has acted [120]. 

The power of jointly inferring demography and selection can be increased by sampling genome data or collecting allele frequencies at several time points (Figure 4b; see, e.g., [12]). An overview on existing methods to analyze such time-sampled data, including the advantages and potential biases, is given in [117]. 

If phenotype data of sequenced individuals are available, it is possible to perform a genome-wide association study (GWAS; Figure 4c), which searches for alleles that are statistically associated with an observed phenotype (e.g., [121]). It should be noted that GWAS studies can yield different effect sizes for a single locus if the genetic structure of the coevolving partner is not taken into account [122]. The authors of this publication further demonstrated the value of integrating genomic information of both co-evolving partners simultaneously into a “two-species co-GWAS” [122]. 

Methods jointly integrating genome data from both coevolving partners are very likely to increase our understanding of the genetic basis underlying coevolution, as the genomes of both partners contain pieces of information on their joint coevolutionary history (Figure 4d). Wang et al. proposed such a method, called Analysis with a Two-Organism Mixed Model or ATOMM, which aims to associate the outcome of reciprocal infection experiments (e.g., the level of quantitative resistance) with genetic variants in the host and parasite genome, simultaneously [123]. Their method also accounts for a latent population structure and allows for different types of genetic variants including insertion/deletion polymorphisms. Nuismer, Jenkins, and Dybdahl proposed a framework to identify coevolutionary loci by measuring the spatial covariation of marker frequencies in the host and parasite across several populations [124]. They showed that the performance of their method mainly depends on the strength of local adaptation between host and parasite, the number of populations being sampled, and the genomic architecture of the trait. 

In summary, we have presented an overview of genetic inference methods. Detecting the genomic basis of coevolution can be achieved by incorporating different sources of information (genotypic information, phenotypic information, information from both interacting partners, or from multiple time points) and combining different sets of methods which are most appropriate for the given system. The overview above also underlines that currently much work is being done on the development and extension of methods that are specifically tailored for the analysis of coevolving systems. 

## 6. Discussion

In this review, we have outlined the evolutionary processes shaping the genomic diversity of viruses and highlighted how they can (individually and interactively) affect genomic diversity (see Figure 3 for an overview). Many analysis tools have been developed by population geneticists to infer the presence and strength of evolutionary forces from genome data of a single species (Figure 4). When applying them, one must be aware of their respective underlying assumptions and to which extent they fit the biology of the virus being studied. Recently developed methods have started to extend beyond a single species analysis framework, enabling the integration of genetic and phenotypic data of coevolving virus and host populations, thus, explicitly taking the reciprocal nature of antagonistic coevolution into account. Such methods provide promising opportunities to identify previously undetected targets of selection, such as resistance genes, virulence factors, or regulatory regions, which will deepen our insights into the molecular basis of host–virus interactions and coevolution. 

When viruses are involved in complex eco-evolutionary feedback loops, species abundances, phenotypic trait distributions, and allele frequencies all change continuously and simultaneously [82]. In such cases, sampling genomic data at several points in time while simultaneously keeping track of phenotype data and population size data enables us to establish links between genetic, phenotypic, and population size changes. Time-sampled genomic data specifically allow for a more precise quantification of the strength of selection [125] and offer more powerful means to disentangle the effects of various ecological and evolutionary forces on genome-wide diversity dynamics [12,117,126]. With their diverse array of life-history traits, life-time strategies, often comparatively small genomes, and short generation times, viruses offer a great opportunity to study the dynamics of complex biological systems in real time. Such a time-resolved multifarious view on ecological and evolutionary dynamics will also increase our mechanistic understanding of the role viruses play in natural ecosystems.

The possibility to sample and analyze data from repeated experiments provides further insights into the diversity of the possible paths antagonistic coevolution can take [97,127] and how the interaction between different kinds of mutations (beneficial, neutral, and deleterious) will shape the resulting eco-evolutionary dynamics [128,129,130]. Analyzing replicated viral genomic data from, e.g., microcosm experiments or different populations with similar environmental properties will allow us to identify conditions under which viral evolution is predictable and will thus aid in understanding and predicting, e.g., disease outbreaks [131]. 

Several challenges remain to be addressed. First, we are not aware of any genomic inference methods that simultaneously take coevolutionary change and ecological population size changes into account. Second, it is important to increase the discussion on optimal sampling schemes (in terms of replication, temporal sampling density, and specific sampling times) to capture as much relevant information as possible in a time- and cost-effective way. Third, there are limits as to how much genomic data can tell us about such highly complex systems, and these limits should be investigated carefully. For all of this, advancement will crucially depend on ongoing exchange between empiricists and theoreticians from various fields, such as virology, ecology, evolutionary biology, and population genetics.

In summary, we have shown in our review how genomic data of viruses—besides helping to delimitate species—offer a powerful source of information to elucidate past ecological and evolutionary processes, to study the genomic basis of adaptation, to improve our understanding of evolution under species interactions, and to shed light on reciprocal interactions between ecological and evolutionary change. These are exciting times in which more and finer-scaled genetic data is increasingly available, and substantial progress is being made in the development of methods linking such data to theory. Both are vital to increase our understanding on how viruses interact with their hosts, how this shapes genomic diversity of both interacting partners, and how this feeds back into ecological processes. 

## Figures and Tables

**Figure 1 viruses-11-00220-f001:**
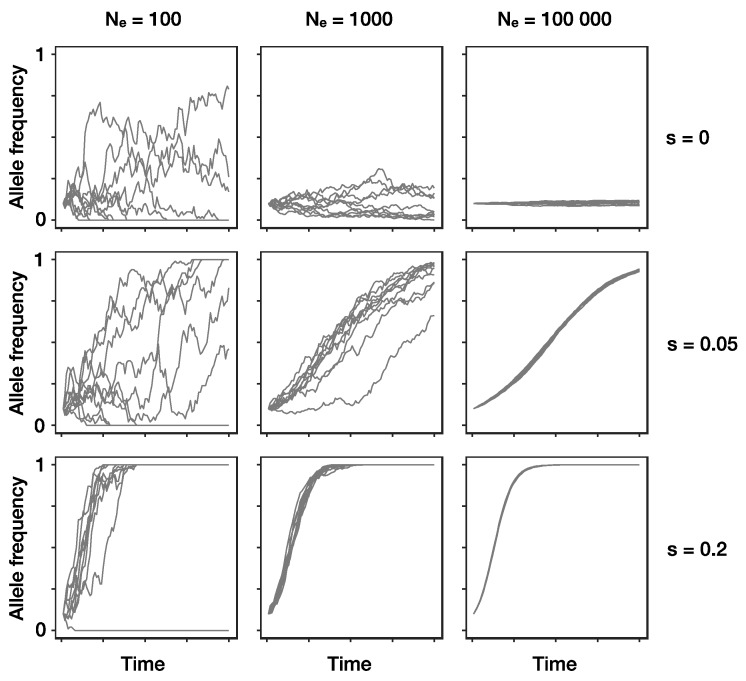
The combined effects of drift and selection on genetic change over time (*x*-axis). Shown are simulated allele frequencies (*y*-axis) of a focal allele for different combinations of effective population size *N_e_* (columns) and selection coefficient *s* (rows). A positive selection coefficient (*s* > 0) indicates a selective advantage of the focal allele compared to the other allele, if *s* = 0 both alleles are neutral and thus, allele frequency changes are only due to genetic drift. Each panel shows the results of ten independent replicates with an initial frequency of 0.1 for the focal allele. Note that when effective population size is small, even positively selected alleles sometimes go extinct due to drift (left column, middle and bottom row). Absolute frequencies *k* of the allele in generation *t + 1* were obtained by randomly drawing from a binomial distribution with $P\left(X=k\right)=\left(\begin{array}{c} N_{e}\\ k \end{array}\right)p_{t+1}^{k}{\left(1-p_{t+1}\right)}^{n-k}$ and $p_{t+1}=\frac{p_{t}\left(1+s\right)}{\bar{w}}$, where *w* denotes the average fitness of the population and *p*_t+1_ denotes the expected frequency of the focal allele without drift in the next generation *t* + 1.

**Figure 2 viruses-11-00220-f002:**
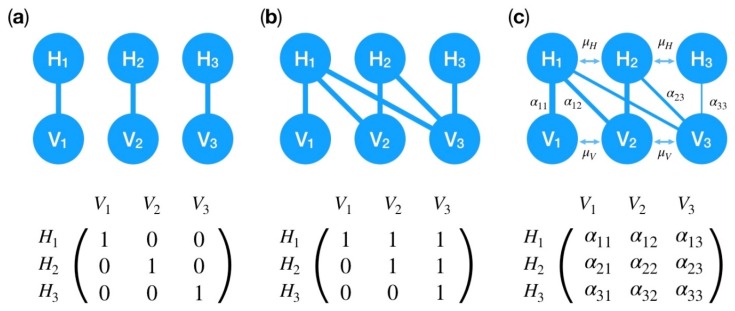
Overview of possible types of coevolutionary interactions between host genotypes (*H_i_*) and virus genotypes (*V_j_*). The top row shows a graphical representation of potential interactions between different host and parasite genotypes. Lines indicate that a virus with genotype *j* can infect a host with genotype *i*. The corresponding infection matrices are shown in the bottom line. Entries in the infection matrix which are equal to *0* (*1*) indicate that the host genotype in row *i* is fully resistant (susceptible) to the virus genotype in column *j*. (**a**) In a matching-allele system each virus genotype can successfully infect only one host genotype. (**b**) In gene-for-gene systems there is one universally infective virus genotype (here V_3_) which is able to infect all host genotypes. Most coevolutionary interactions fall onto a continuum between these two extremes and can be captured in a correspondingly parameterized infection matrix as illustrated in (**c**). *α_ij_* reflects the rate of success for virus genotype *j* to infect host genotype *i*. Every *α_ij_* can take values between *0* and *1*. Genotype-altering mutations happen at rate µ_H_ in the host and µ_P_ in the parasite.

**Figure 3 viruses-11-00220-f003:**
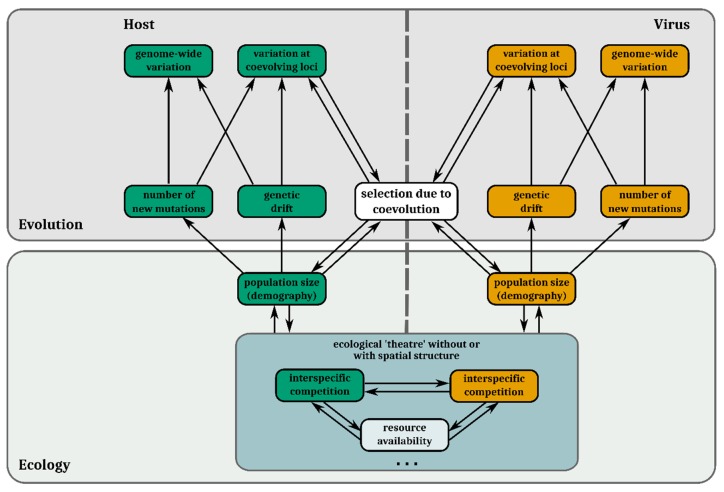
Potential interactions between ecological processes and evolutionary processes in host–virus coevolution and how they interact with and shape genomic variation. In the evolution section, boxes in the top row correspond to genomic variation and those in the second row (bottom) to evolutionary forces governing them. Eco-evolutionary feedback loops take place when coevolutionary selection alters population sizes and/or other ecological processes (third row and below), which in turn alters how the different genomic forces affect genomic variation. The dots at the bottom indicate that the above-mentioned ecological processes by no means constitute an exhaustive list. Features of the host are always presented in green and features of the virus in orange. Selection imposed by abiotic variables is not included in this figure.

**Figure 4 viruses-11-00220-f004:**
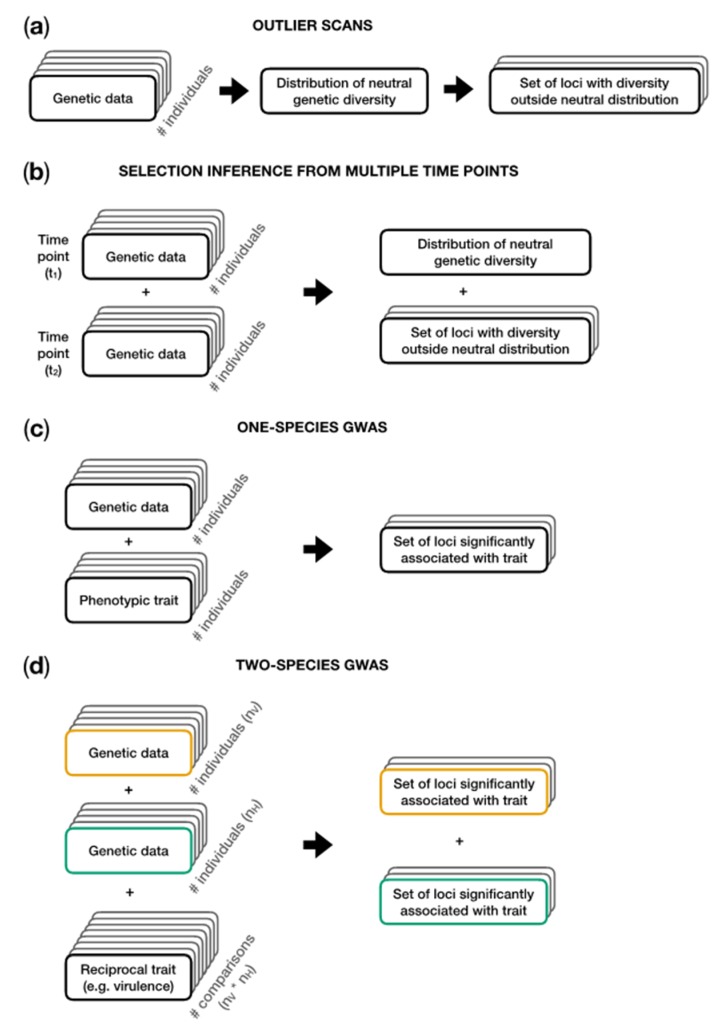
Analysis steps involved in the genetic inference methods outlined in Section 5. (**a**) In outlier scans, genetic data are used to obtain an estimate of the demographic history and the distribution of neutral diversity given this demography. Loci which are at the extremes or even outside of this neutral distribution are subsequently identified as putatively under selection. (**b**) Genetic data from multiple time points allow the calculation of changes in allele frequencies over time. This increases the power to jointly estimate the demography and identify loci under selection. (**c**) Genome-wide association studies (GWAS) are performed with phenotypic and genetic information from a sample of individuals within a population to detect associations between genetic variants and a certain phenotype (e.g., quantitative virulence). (**d**) Two-species GWAS integrates genomic information from a sample of n_H_ host individuals and n_V_ virus individuals and phenotypic outcome of all n_H_ * n_V_ pairwise interactions. Data from the virus are shown in yellow. Data from the host are shown in green.

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
