# Peer review of "Ecological and Evolutionary Processes Shaping Viral Genetic Diversity"

_viruses, 2019, doi:10.3390/v11030220_

Reviewer 1 Report

Detailed review is attached as a PDF file

Author Response

Ecological and evolutionary processes shaping viral genetic diversity

Response to Reviewer 1

The manuscript is written well, and covers the majority of literature on the subject. However,there are several aspects that I believe need to be improved before this review is accepted for publication.

We would like to thank the reviewer for the positive and useful feedback. The outside perspective has aptly identified some aspects that have (consciously or unconsciously) been neglected in the manuscript so far.

One of the aspects that the authors need to clarify is the type of viruses that they refer to. In other words a mention is made about the mutation rate of DNA and RNA viruses however is this also relevant for other viruses that are not small and likely have a much smaller mutation rate? Recent research in the last 15 years has revealed that some viruses of amoeba can have giant genomes encoding for hundreds and thousands, including many interesting metabolic pathways (e.g. megaviruses, tupanviruses, mimiviruses,etc.). Other giant dsDNAviruses infect microalgae. Very little is known of the evolutionary processes of these viruses and it is likely that the some of what is described in the current review does not apply to these. This needs to be made clear in the review.

The reviewer justly points out that recent discoveries of several groups of giant viruses have greatly stretched our expectations of the range of molecular genomic characteristics possessed by viruses as a whole (in terms of among other things genome size, number of protein-coding regions, mutation rate). However, from a 'general' population genetics perspective, even giant viruses of amoebae have small genomes, high mutation rates, short generation times and large population sizes, as do most other microorganisms. In line with this, we do not see a need to exclude giant viruses from our review, or discuss them separately. We now explicitly state that we do not focus on any particular group of virus in the introduction.

The reviewer also notes that we currently know little about how these giant viruses evolve and how genetically stable lineages are. Hopefully our review can be a useful tool when thinking about how evolution of giant viruses might differ from other viruses.

 Another aspect that is currently lacking from the review is examples of specifics. In other words a lot is referred to certain systems, but specific in depth examples of certain hosts and their viruses is not given in detail. Rather it is left for the reader to chase up the quoted references. I think that in such a review some specific examples of where the methodology and approach reviewed has been applied should be given.

We added examples for observed burst sizes and observed genetic variation within virus populations (section 2: Viral population genetics).

In addition, I suggest that the material written in Box1 is moved out of the box and listed as a separate paragraph. I also could not see the entire box in this version of the manuscript and there was a lot of information missing form that version. This was probably due to formatting or uploading the document. Nevertheless,this needs fixing.

We apologise for the formatting error that made the last lines of the first half of Box 1 invisible. Box 1 has now been integrated into the main text as section 5: Genomic inference methods.

Finally, I think that a discussion of the roles the environment plays on host-virus evolution or the pressure it poses on viruses was lacking. We now know that with temperate and lysogenic viruses, environmental conditions are crucial in triggering a shift from an integrated stage within the host genome to a lytic stage. Some discussion is warranted on this or at least it needs to be mentioned somewhere.

We acknowledge the effect of abiotic environmental factors on viruses, and that these are largely neglected in our review. The described scope of the review (section 1: Introduction) now has been altered to explicitly state this.

 Specific comments:

“time sampled data” written in the box needs to be explained or rewritten, it is not the correct term to use.

By time-sampled data we mean samples from a population taken at different points in time, sensu e.g. [1]. We now introduce the term before its first use in section 5: Genomic inference methods at lines 340-343.

Line 201-205 This is a very long sentence. Consider breaking it up and splitting to two. Also, note that Chlorovirus is a giant virus. Consider a different example.

We broke the sentence up into three. We do however generally not see the need to refrain from using giant viruses as examples in this review (see also our response above to the comment about giant viruses of amoebae). Besides, we are not aware of any other host-virus systems where the presence of a full feedback loop between ecological and evolutionary change has been causally established.

All other formulation comments were addressed, as visible by Word TrackChanges in the resubmission.

References

1.        Foll, M.; Poh, Y.P.; Renzette, N.; Ferrer-Admetlla, A.; Bank, C.; Shim, H.; Malaspinas, A.S.; Ewing, G.; Liu, P.; Wegmann, D.; et al. Influenza virus drug resistance: A time-sampled population genetics perspective. PLoS Genet. 2014, 10.

Reviewer 2 Report

The manuscript is a very interesting and well-written review on factors affecting the dynamics and evolution of virus populations and is appropriate for publication in Viruses following minor modifications.

Comments:

It is unclear to me whether this manuscript is supposed to focus on bacterial viruses. If yes, I would encourage the authors to use more examples derived from e.g. bacterial host - bacteriophage interactions, whenever possible.

I am not sure about "Box 1". Maybe this could become another section of the manuscript and the authors could prepare a diagram illustrating the steps of the analysis they describe in detail in the text.

Lines 81-88: Maybe the authors could properly introduce negative/purifying selection here?

Author Response

Ecological and evolutionary processes shaping viral genetic diversity

Response to Reviewer 2

The manuscript is a very interesting and well-written review on factors affecting the dynamics and evolution of virus populations and is appropriate for publication in Viruses following minor modifications.

We would like to thank the reviewer for the positive feedback.

Comments:

It is unclear to me whether this manuscript is supposed to focus on bacterial viruses. If yes, I would encourage the authors to use more examples derived from e.g. bacterial host - bacteriophage interactions, whenever possible.

We always intended to not confine ourselves to microbial host-virus systems, but instead to also draw upon literature from systems with multicellular hosts. Such systems have contributed essential knowledge about antagonistic (eco-)coevolutionary interactions. We now explicitly state that our review is not confined to specific viral groups in section 1: Introduction.

I am not sure about "Box 1". Maybe this could become another section of the manuscript and the authors could prepare a diagram illustrating the steps of the analysis they describe in detail in the text.

Box 1 has now been integrated into the main text as section 5: Genomic inference methods. Additionally, figure 4 was added. It depicts the steps involved in genetic data analysis, and how this changes with the availability of information from multiple time points, multiple species, and/or on phenotypic traits.

Lines 81-88: Maybe the authors could properly introduce negative/purifying selection here?

We now introduce the term ‘purifying selection’ next to positive selection, and we rephrased the background selection sentence.
